# JEN-1: Text-Guided Universal Music Generation with Omnidirectional Diffusion Models

## Abstract

Music generation has attracted growing interest with the advancement of deep generative models. However, generating music conditioned on textual descriptions, known as text-to-music, remains challenging due to the complexity of musical structures and high sampling rate requirements. Despite the task's significance, prevailing generative models exhibit limitations in music quality, computational efficiency, and generalization ability. This paper introduces **JEN-1**, a universal high-fidelity model for text-to-music generation. JEN-1 is a diffusion model incorporating both autoregressive and non-autoregressive training in an end-to-end manner, enabling up to 48kHz high-fidelity stereo music generation. Through multi-task in-context learning, JEN-1 performs various generation tasks including text-guided music generation, music inpainting, and continuation. Evaluations demonstrate JEN-1's superior performance over state-of-the-art methods in text-music alignment and music quality while maintaining computational efficiency. Our anonymous demo pages are available at `https://anonymous.4open.science/w/Jen1-Demo-Page-21D4/`

*"Music is the universal language of mankind."*

*– Henry Wadsworth Longfellow*

## 1 Introduction

Music, as an artistic expression comprising harmony, melody and rhythm, holds great cultural significance and appeal to humans. Recent years have witnessed remarkable progress in music generation with the rise of deep generative models (Liu et al., 2023; Kreuk et al., 2022; Agostinelli et al., 2023). However, generating high-fidelity and realistic music still poses unique challenges compared to other modalities. Firstly, music utilizes the full frequency spectrum, requiring high sampling rates like 44.1kHz stereo to capture the intricacies. This is in contrast to speech generation which focuses on linguistic content and uses lower sampling rates (*e.g.*, 16kHz). Secondly, the blend of multiple instruments and the arrangement of melodies and harmonies result in highly complex structures. With humans being sensitive to musical dissonance, music generation allows little room for imperfections. Most critically, controllability over attributes like key, genre and melody is indispensable for creators to realize their artistic vision.

The multi-modal intersection of text and music, known as text-to-music generation, offers valuable capabilities to bridge free-form textual descriptions and musical compositions. A variety of works have been done towards text-to-music generation. Regarding how they encode the high complexity of input music signals, typical methods are generally categorized into, (1) directly adopting waveform signals with the help of discrete encodings to handle the complexity burden (Agostinelli et al., 2023; Copet et al., 2023), or (2) converting signals to spectrum formats that allow continuous encodings, which, inevitably incur fidelity losses during the spectrum conversion process (Liu et al., 2023; Ghosal et al., 2023). Apart from these works, Noise2Music (Huang et al., 2023a) tries to leverage waveform signals with continuous encodings yet the fidelity of music representation is bottle-necked to only 3.2kHz, which is still limited even after adding a cascade model and a super-resolution stage hierarchically.

Table 1: Comparison between state-of-the-art music generative models.

| | Feature | MusicLM | MusicGen | AudioLDM | Noise2Music | **JEN-1 (Ours)** |
|---|---|---|---|---|---|---|
| Data | high sample rate | ✗ | ✗ | ✗ | ✗ | ✓ |
| | 2-channel stereo | ✗ | ✗ | ✗ | ✗ | ✓ |
| | waveform | ✓ | ✓ | ✗ | ✓ | ✓ |
| Model | autoregressive | ✓ | ✓ | ✗ | ✗ | ✓ |
| | non-autoregressive | ✗ | ✗ | ✓ | ✓ | ✓ |
| | non-cascade model | ✗ | ✓ | ✓ | ✗ | ✓ |
| Task | single-task training | ✓ | ✓ | ✓ | ✓ | ✓ |
| | multi-task training | ✗ | ✗ | ✗ | ✗ | ✓ |

A systematic comparison of existing methods is provided in Table 1, where some of the approaches operate on spectrogram representations, incurring fidelity loss from audio conversion (Liu et al., 2023; Ghosal et al., 2023). Others employ inefficient autoregressive designs to sequentially predict each music token (Agostinelli et al., 2023), or adopt multiple models to cascade-generate the music, resulting in low computational efficiency (Copet et al., 2023; Huang et al., 2023a). More restrictively, the training objectives of existing methods are confined to a single task, lacking the versatility to conduct multiple and general music generation and editing tasks.

In this paper, we propose **JEN-1**, a universal text-to-music generation model combining quality, controllability, and efficiency. JEN-1 leverages a hybrid autoregressive (AR) and non-autoregressive (NAR) structure, with a novel omnidirectional design that potentially enjoys the benefits of both generation efficiency and quality. By the combination of AR and NAR structural designs, for the first time, JEN-1 unifies the learning process of multiple tasks including text-to-music, music inpainting, and music continuation into one single model. Another hallmark of JEN-1 is the ability to encode raw waveform data formats, *e.g.*, enabling the generation of high-fidelity 48kHz stereo audios, which is realized by our proposed masked noise-robust autoencoder with specific designs for continuous embedding representation and latent embedding normalization.

We extensively evaluate JEN-1 against state-of-the-art baselines across objective metrics and human evaluations. Results indicate that JEN-1 produces music of perceptually higher quality compared to the current best methods (85.7 *vs.* 83.8). Ablations validate the efficacy of each technical component and also demonstrate that JEN-1 benefits greatly from multitask joint training. More importantly, human judges confirm JEN-1 generates music highly aligned with text prompts in a melodically and harmonically pleasing fashion.

In summary, the key contributions of this work are:

1. We propose JEN-1 as a universal framework to the challenging text-to-music generation task. JEN-1 utilizes an extremely efficient approach by directly modeling waveforms and avoids the conversion loss associated with spectrograms. It incorporates a masked autoencoder and diffusion model, yielding high-quality music at a 48kHz sampling rate.

2. JEN-1 integrates both autoregressive and non-autoregressive diffusion modes in an end-to-end manner to improve sequential dependency and enhance sequence generation concurrently, enabling music generation, music continuation, and music inpainting within one single non-cascade model.

3. We conduct comprehensive evaluations, both objective and involving human judgment, to thoroughly assess the crucial design choices underlying our method. Results demonstrate that JEN-1 generates melodically aligned music that adheres to textual descriptions while maintaining high fidelity.

## 2 RELATED WORK

In this section, we present a review of the extant literature within the domain of music generation. We undertake a comparative analysis of three primary paradigmatic distinctions: single-task *vs.* multi-task training, waveform *vs.* spectrum-based methods, and autoregressive *vs.* non-autoregressive generative modes.

**Single-task *vs.* Multi-task.** Several prior symbolic music generation methods Yang et al. (2017); Sulun et al. (2022) have primarily produced basic MIDI-style compositions, characterized by their

simplicity and lack of realism. In contrast, recent methodologies Agostinelli et al. (2023); Liu et al. (2023); Huang et al. (2023a) have employed text descriptions as conditioning inputs to facilitate the direct generation of high-fidelity raw musical compositions. Nevertheless, existing approaches are often constrained by narrow generation objectives, limiting their applicability to a singular task and impeding their adaptability for diverse music generation and editing tasks. In contrast, JEN-1 introduces a universal framework designed to tackle the intricate text-to-music generation challenge. This framework streamlines the learning process by encompassing multiple tasks within a singular mode, thus encompassing text-to-music conversion, music inpainting, and music continuation.

**Waveform *vs*. Spectrum.** The fundamental processes of audio feature extraction and representation learning play a pivotal role in music generation and can be classified into two main directions. One of these methods entails the transformation of the audio waveform into a mel-spectrogram Liu et al. (2023); Ghosal et al. (2023), which is then processed as images. However, this transformation from waveform to spectrogram necessitates a shift from continuous encodings to spectral formats, a step that inevitably introduces fidelity losses in the audio data. Conversely, the other approaches Agostinelli et al. (2023); Huang et al. (2023a) involve the direct utilization of raw waveform-based audio data. Some variations of this approach go a step further by converting the audio into discrete tokens Agostinelli et al. (2023); Copet et al. (2023) for modeling purposes, also resulting in a compromise in audio quality. In this study, JEN-1 adopts a strategy that preserves the raw waveform data in a continuous space format, facilitating the generation of high-fidelity stereo audio at a sampling rate of 48kHz.

**Autoregressive *vs*. Non-autoregressive.** Music generation draws inspiration from both natural language processing (NLP) and computer vision (CV), incorporating both autoregressive (AR) and non-autoregressive (NAR) models. AR models, such as PerceiverAR Hawthorne et al. (2022), AudioGen Kreuk et al. (2022), MusicLM Agostinelli et al. (2023), and Jukebox Dhariwal et al. (2020), predict audio tokens sequentially based on prior context. However, their processing speed limitations hinder their utility in music in-painting tasks. Conversely, NAR models generate multiple tokens concurrently, offering faster processing and improved generative performance. Modern NAR models, such as StableDiffusion Rombach et al. (2022), excel in image generation, while diffusion models Ho et al. (2020) have gained attention for their NAR generative capabilities in music generation tasks. These models progressively refine random noise into latent representations, enabling efficient synthesis of high-quality audio. Some approaches, including Make-An-Audio Huang et al. (2023b), Noise2Music Huang et al. (2023a), AudioLDM Liu et al. (2023), and TANGO Ghosal et al. (2023), extend latent diffusion models (LDM) Rombach et al. (2022) and demonstrate significant speed advantages in music generation tasks. In this study, JEN-1 pioneers a hybrid AR and NAR structure to enhance sequential dependency and improve concurrent sequence generation.

## 3 PRELIMINARY

### 3.1 CONDITIONAL GENERATIVE MODELS

In the field of content synthesis, the implementation of conditional generative models often involves applying either autoregressive (AR) (Agostinelli et al., 2023; Copet et al., 2023) or non-autoregressive (NAR) (Liu et al., 2023; Ghosal et al., 2023) paradigms. The inherent structure of language, where each word functions as a distinct token and sentences are sequentially constructed from these tokens, makes the AR paradigm a more natural choice for language modeling. Thus, in the domain of Natural Language Processing (NLP), transformer-based models, *e.g.*, GPT series, have emerged as the prevailing approach for text generation tasks. AR methods (Agostinelli et al., 2023; Copet et al., 2023) rely on predicting future tokens based on visible history tokens. The likelihood is represented by:

$$p_{\mathrm{AR}}(\boldsymbol{y} \mid \boldsymbol{x}) = \prod_{i=1}^{N} p\left(\boldsymbol{y}_i \mid \boldsymbol{y}_{1:i-1}; \boldsymbol{x}\right), \qquad (1)$$

where $\boldsymbol{y}_i$ represents the $i$-th token in sequence $\boldsymbol{y}$.

Conversely, for the image generation task where images have no explicit time series structure and images typically occupy continuous space, employing an NAR approach is deemed more suitable. Notably, the NAR approach, such as stable diffusion, has emerged as the dominant method for

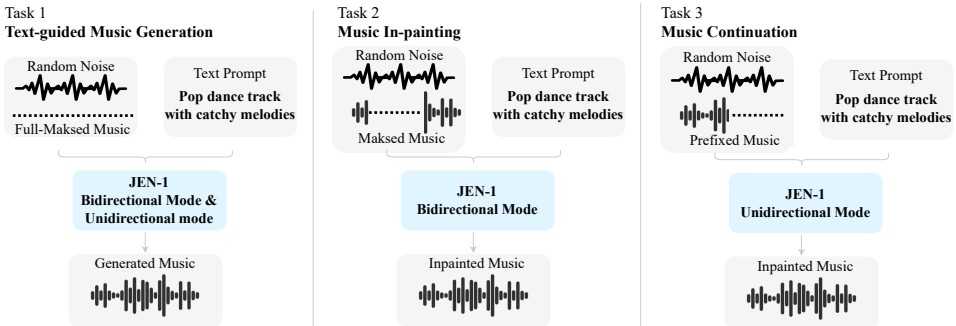

Figure 1: Illustration of the JEN-1 multi-task training strategy, including the text-guided music generation task, the music inpainting task, and the music continuation task. JEN-1 achieves the in-context learning task generalization by concatenating the noise and masked audio in a channel-wise manner. JEN-1 integrates both the bidirectional mode to gather comprehensive context and the unidirectional mode to capture sequential dependency.

addressing image generation tasks. NAR approaches assume conditional independence among latent embeddings and generate them uniformly without distinction during prediction. This results in a likelihood expressed as:

$$p_{\text{NAR}}(\boldsymbol{y} \mid \boldsymbol{x}) = \prod_{i=1}^{N} p\left(\boldsymbol{y}_i \mid \boldsymbol{x}\right). \tag{2}$$

Although the parallel generation approach of NAR offers a notable speed advantage, it falls short in terms of capturing long-term consistency.

In this work, we argue that audio data can be regarded as a hybrid form of data. It exhibits characteristics akin to images, as it resides within a continuous space that enables the modeling of high-quality music. Additionally, audio shares similarities with text in its nature as a time-series data. Consequently, we propose a novel approach in our JEN-1 design, which entails the amalgamation of both the auto-regressive and non-autoregressive modes into a cohesive omnidirectional diffusion model.

## 3.2 DIFFUSION MODELS FOR AUDIO GENERATION

Diffusion models (Ho et al., 2020) constitute probabilistic models explicitly developed for the purpose of learning a data distribution $p(\boldsymbol{x})$. The overall learning of diffusion models involves a forward *diffusion* process and a gradual *denoising* process, each consisting of a sequence of $T$ steps that act as a Markov Chain. In the forward diffusion process, a fixed linear Gaussian model is employed to gradually perturb the initial random variable $\boldsymbol{z}_0$ until it converges to the standard Gaussian distribution. This process can be formally articulated as follows,

$$q\left(\boldsymbol{z}_t \mid \boldsymbol{z}_0; \boldsymbol{x}\right) = \mathcal{N}\left(\boldsymbol{z}_t; \sqrt{\bar{\alpha}_t}\boldsymbol{z}_0, \left(1 - \bar{\alpha}_t\right)\mathbf{I}\right),$$
$$\bar{\alpha}_t = \prod_{i=1}^{t} \alpha_i, \tag{3}$$

where $\alpha_i$ is a coefficient that monotonically decreases with timestep $t$, and $\boldsymbol{z}_t$ is the latent state at timestep $t$. The reverse process is to initiate from standard Gaussian noise and progressively utilize the denoising transition $p_{\boldsymbol{\theta}}\left(\boldsymbol{z}_{t-1} \mid \boldsymbol{z}_t; \boldsymbol{x}\right)$ for generation,

$$p_{\boldsymbol{\theta}}\left(\boldsymbol{z}_{t-1} \mid \boldsymbol{z}_t; \boldsymbol{x}\right) = \mathcal{N}\left(\boldsymbol{z}_{t-1}; \mu_{\boldsymbol{\theta}}\left(\boldsymbol{z}_t, t; \boldsymbol{x}\right), \Sigma_{\boldsymbol{\theta}}\left(\boldsymbol{z}_t, t; \boldsymbol{x}\right)\right), \tag{4}$$

where the mean $\mu_{\boldsymbol{\theta}}$ and variance $\Sigma_{\boldsymbol{\theta}}$ are learned from the model parameterized by $\theta$. We use predefined variance without trainable parameters following (Rombach et al., 2022; Liu et al., 2023). After simply expanding and re-parameterizing, our training objective of the conditional diffusion model can be denoted as,

$$\mathcal{L} = \mathbb{E}_{\boldsymbol{z}_0, \epsilon \sim \mathcal{N}(0,1), t}\left[\|\epsilon - \epsilon_\theta\left(\boldsymbol{z}_t, t\right)\|_2^2\right], \tag{5}$$

where $t$ is uniformly sampled from $\{1, ..., T\}$, $\epsilon$ is the ground truth of the sampled noise, and $\epsilon_\theta(\cdot)$ is the noise predicted by the diffusion model.

The conventional diffusion model is characterized as a non-autoregressive model, which poses challenges in effectively capturing sequential dependencies in music flow. To address this limitation, we propose the joint omnidirectional diffusion model JEN-1, an integrated framework that leverages both unidirectional and bidirectional training. These adaptations allow for precise control over the contextual information used to condition predictions, enhancing the model's ability to capture sequential dependencies in music data.

## 4 METHOD

In this research paper, we propose a novel model called JEN-1, which utilizes an omnidirectional 1D diffusion model. JEN-1 combines bidirectional and unidirectional modes, offering a unified approach for universal music generation conditioned on either text or music inputs. The model operates in a noise-robust latent embedding space obtained from a masked audio autoencoder, enabling high-fidelity reconstruction from latent embeddings with a low frame rate(§ 4.1). In contrast to prior generation models that use discrete tokens or involve multiple cascaded stages, JEN-1 introduces a unique modeling framework capable of generating continuous, high-fidelity music using one single model. JEN-1 effectively utilizes both autoregressive training to improve sequential dependency and non-autoregressive training to enhance sequence generation concurrently (§ 4.2). By employing in-context learning and multi-task learning, one of the significant advantages of JEN-1 is its support for conditional generation based on either text or melody, enhancing its adaptability to various creative scenarios (§ 4.3). This flexibility allows the model to be applied to different music generation tasks, making it a versatile and powerful tool for music composition and production.

### 4.1 MASKED AUTOENCODER FOR HIGH FIDELITY LATENT REPRESENTATION LEARNING

**High Fidelity Neural Audio Latent Representation.** To facilitate the training on limited computational resources without compromising quality and fidelity, our approach JEN-1 employs a high-fidelity audio autoencoder $\mathcal{E}$ to compress original audio into latent representations $z$. Formally, given a two-channel stereo audio $x \in \mathbb{R}^{L \times 2}$, the encoder $\mathcal{E}$ encodes $x$ into a latent representation $z = \mathcal{E}(x)$, where $z \in \mathbb{R}^{L/h \times c}$. $L$ is the sequence length of given music, $h$ is the hop size and $c$ is the dimension of latent embedding. While the decoder reconstructs the audio $\tilde{x} = \mathcal{D}(z) = \mathcal{D}(\mathcal{E}(x))$ from the latent representation. Our audio compression model is inspired and modified based on previous work (Zeghidour et al., 2021; Défossez et al., 2022), which consists of an autoencoder trained by a combination of a reconstruction loss over both time and frequency domains and a patch-based adversarial objective operating at different resolutions. This ensures that the audio reconstructions are confined to the original audio manifold by enforcing local realism and avoids muffled effects introduced by relying solely on sample-space losses with L1 or L2 objectives. Unlike prior endeavors (Zeghidour et al., 2021; Défossez et al., 2022) that employ a quantization layer to produce the discrete codes, our model directly extracts the continuous embeddings without any quality-reducing loss due to quantization. This utilization of powerful autoencoder representations enables us to achieve a nearly optimal balance between complexity reduction and high-frequency detail preservation, leading to a significant improvement in music fidelity.

**Noise-robust Masked Autoencoder.** To further enhance the robustness of decoder $\mathcal{D}$, we propose a masking strategy, which effectively reduces noises and mitigates artifacts, yielding superior-quality audio reconstruction. In our training procedure, we adopt a specific technique wherein $p = 5\%$ of the intermediate latent embeddings are randomly masked before feeding into the decoder. By doing so, we enable the decoder to acquire proficiency in reconstructing superior-quality data even when exposed to corrupted inputs. We train the autoencoder on 48kHz stereophonic audios with large batch size and employ an exponential moving average to aggregate the weights. As a result of these enhancements, the performance of our audio autoencoder surpasses that of the original model in all evaluated reconstruction metrics. Consequently, we adopt this audio autoencoder for all of our subsequent experiments.

**Normalizing Latent Embedding Space.** To avoid arbitrarily scaled latent spaces, (Rombach et al., 2022) found it is crucial to achieve better performance by estimating the component-wise variance and re-scale the latent $z$ to have a unit standard deviation. In contrast to previous approaches that only estimate the component-wise variance, JEN-1 employs a straightforward yet effective post-processing technique to address the challenge of anisotropy in latent embeddings as shown in Algo-

---

**Algorithm 1** Normalizing Latent Embedding Space

---

**Input:** Existing latent embeddings $\{z_i\}_{i=1}^N$ and reduced dimension $k$

1: compute $\mu$ and $\Sigma$ of $\{z_i\}_{i=1}^N$
2: compute $U, \Lambda, U^T = \mathbf{SVD}(\Sigma)$
3: compute $W = (U\sqrt{\Lambda^{-1}})[:,:k]$
4: $\widetilde{z}_i = (z_i - \mu)W$

**Output:** Normalized latent embeddings $\{\widetilde{z}_i\}_{i=1}^N$

---

rithm 1. Specially, we channel-wisely perform zero-mean normalization on the latent embedding, and then transform the covariance matrix to the identity matrix via Singular Value Decomposition (SVD) algorithm. We implement a batch-incremental equivalent algorithm to calculate these transformation statistics. Additionally, we incorporate a dimension reduction strategy with 5% least important channels to enhance the whitening process further and improve the overall effectiveness of our approach.

## 4.2 OMNIDIRECTIONAL LATENT DIFFUSION MODELS

In some prior approaches (Liu et al., 2023; Ghosal et al., 2023), time-frequency conversion techniques, such as mel-spectrogram, have been employed for transforming the audio generation into an image generation problem. Nevertheless, we contend that this conversion from raw audio data to mel-spectrogram inevitably leads to a significant reduction in quality. To address this concern, JEN-1 directly leverages a temporal 1D efficient U-Net. This modified version of the Efficient U-Net (Saharia et al., 2022) allows us to effectively model the waveform and implement the required blocks in the diffusion model. The U-Net model's architecture comprises cascading down-sampling and up-sampling blocks interconnected via residual connections. Each down/up-sampling block consists of a down/upsampling layer, followed by a set of blocks that involve 1D temporal convolutional layers, and self/cross-attention layers. Both the stacked input and output are represented as latent sequences of length $L$, while

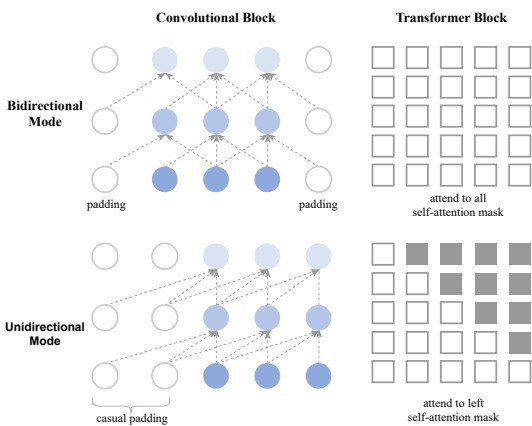

Figure 2: Illustration of bidirectional mode and unidirectional mode for convolutional block and transformer block. In the unidirectional mode, we use causal padding in the convolutional block and attend the self-attention mask only to the left context in the transformer block.

the diffusion time $t$ is encoded as a single-time embedding vector that interacts with the model via the aforementioned combined layers within the down and up-sampling blocks. In the context of the U-Net model, the input consists of the noisy sample denoted as $x_t$, which is stacked with additional conditional information. The resulting output corresponds to the noise prediction $\epsilon$ during the diffusion process.

**Task Generalization via In-context Learning.** To better achieve the goal of multi-task versatility, we propose a novel omnidirectional latent diffusion model without explicitly changing the U-Net architecture. As shown in Figure 1, JEN-1 formulates various music generation tasks as text-guided in-context learning tasks. The common goal of these in-context learning tasks is to produce diverse and realistic music that is coherent with the context music and has the correct style described by the text. For in-context learning objectives, *e.g.*, music inpainting task, and music continuation task, additional masked music information, which the model is conditioned upon, can be extracted into latent embedding and stacked as additional channels in the input. More precisely, apart from the original latent channels, the U-Net block has 129 additional input channels (128 for the encoded masked audio and 1 for the mask itself).

**From Bidirectional Mode to Unidirectional Mode.** To account for the inherent sequential characteristic of music, JEN-1 integrates the unidirectional diffusion mode by ensuring that the generation of latent on the right depends on the generated ones on the left, a mechanism achieved through employing a unidirectional self-attention mask and a causal padding mode in convolutional blocks. In general, the architecture of the omnidirectional diffusion model enables various input pathways, facilitating the integration of different types of data into the model, resulting in versatile and powerful capabilities for noise prediction and diffusion modeling. During training, JEN-1 could switch between a unidirectional mode and a bidirectional model without changing the architecture of the model. The parameter weight is shared for different learning objectives. As illustrated in Figure 2, JEN-1 could switch into the unidirectional (autoregressive) mode, *i.e.*, the output variable depends only on its own previous values. We employ *causal padding* (Oord et al., 2016) in all 1D convolutional layers, padding with zeros in the front so that we can also predict the values of early time steps in the frame. In addition, we employ a triangular attention mask following (Vaswani et al., 2017), by padding and masking future tokens in the input received by the self-attention blocks.

### 4.3 UNIFIED MUSIC MULTI-TASK TRAINING

In contrast to prior methods that solely rely on a single text-guided learning objective, our proposed framework, JEN-1, adopts a novel approach by simultaneously incorporating multiple generative learning objectives while sharing common parameters. As depicted in Figure 1, the training process encompasses three distinct music generation tasks: bidirectional text-guided music generation, bidirectional music inpainting, and unidirectional music continuation. The utilization of multi-task training is a notable aspect of our approach, allowing for a cohesive and unified training procedure across all desired music generation tasks. This approach enhances the model's ability to generalize across tasks, while also improving the handling of music sequential dependencies and the concurrent generation of sequences.

**Text-guided Music Generation Task.** In this task, we employ both the bidirectional and unidirectional modes. The bidirectional model allows all latent embeddings to attend to one another during the denoising process, thereby enabling the encoding of comprehensive contextual information from both preceding and succeeding directions. On the other hand, the unidirectional model restricts all latent embeddings to attend solely to their previous time counterparts, which facilitates the learning of temporal dependencies in music data. Moreover, for the purpose of preserving task consistency within the framework of U-Net stacked inputs, we concatenate a full-size mask alongside all-empty masked audio as the additional condition.

**Music inpainting Task.** In the domain of audio editing, inpainting denotes the process of restoring missing segments within the music. This restorative technique is predominantly employed to reconstruct corrupted audio from the past, as well as to eliminate undesired elements like noise and watermarks from musical compositions. In this task, we adopt the bidirectional mode in JEN-1. During the training phase, our approach involves simulating the music inpainting process by randomly generating audio masks with mask ratios ranging from 20% to 80%. These masks are then utilized to obtain the corresponding masked audio, which serves as the conditional in-context learning inputs within the U-Net model.

**Music Continuation Task.** We demonstrate that the proposed JEN-1 model facilitates both music inpainting (interpolation) and music continuation (extrapolation) by employing the novel omnidirectional diffusion model. The conventional diffusion model, due to its non-autoregressive nature, has demonstrated suboptimal performance in previous studies (Borsos et al., 2023; Agostinelli et al., 2023). This limitation has impeded its successful application in audio continuation tasks. To address this issue, we adopt the unidirectional mode in our music continuation task, ensuring that the predicted latent embeddings exclusively attend to their leftward context within the target segment. Similarly, we simulate the music continuation process through the random generation of exclusive right-only masks. These masks are generated with varying ratios spanning from 20% to 80%.

The overall training objective is the sum of the different types of JEN-1 tasks described above. Specifically, within a training batch, for 1/3 of the data we use the bidirectional text-guided generation objective, for 1/3 of the data we employ the bidirectional music inpainting objective, and the unidirectional music continuation objective is calculated with the rest of 1/3 data. To gain a more comprehensive overview of JEN-1, we kindly direct your attention to our demo page.

Table 2: Comparison with state-of-the-art text-to-music generation methods on MusicCaps test set.

| | QUANTITATIVE | | | QUALITATIVE | | EFFICIENCY |
|---|---|---|---|---|---|---|
| METHODS | FAD↓ | KL↓ | CLAP↑ | T2M-QLT↑ | T2M-ALI↑ | PARAMETERS↓ |
| Riffusion | 14.8 | 2.06 | 0.19 | 72.1 | 72.2 | 890M |
| Mousai | 7.5 | 1.59 | 0.23 | 76.3 | 71.9 | 857M |
| MusicLM | 4.0 | - | - | 81.7 | 82.0 | 860M |
| Noise2Music | 2.1 | - | - | - | - | 1.3B |
| MusicGen | 3.8 | **1.22** | 0.31 | 83.8 | 79.5 | 3.3B |
| **JEN-1 (Ours)** | **2.0** | 1.29 | **0.33** | **85.7** | **82.8** | **746M** |

## 5 EXPERIMENT

**Implementation Details.** For the masked music autoencoder, we used a hop size of 320, resulting in 125Hz latent sequences for encoding 48kHz music audio. The dimension of latent embedding is 128. We randomly mask 5% of the latent embedding during training to achieve a noise-tolerant decoder. We employ FLAN-T5 (Chung et al., 2022), an instruct-based large language model to provide superior text embedding extraction. For the omnidirectional diffusion model, we set the intermediate cross-attention dimension to 1024, resulting in 746M parameters. During the multi-task training, we evenly allocate 1/3 of a batch to each training task. In addition, we applied the classifier-free guidance (Ho & Salimans, 2022) to improve the correspondence between samples and text conditions. During training, the cross-attention layer is randomly replaced by self-attention with a probability of 0.2. We train our JEN-1 models on 8 A100 GPUs for 200k steps with the AdamW optimizer (Loshchilov & Hutter, 2017), a linear-decayed learning rate starting from $3e^{-5}$ a total batch size of 512 examples, $\beta_1 = 0.9$, $\beta_2 = 0.95$, a decoupled weight decay of 0.1, and gradient clipping of 1.0.

**Datasets.** We use total 5k hours of high-quality private music data to train JEN-1. All music data consist of full-length music sampled at 48kHz with metadata composed of a rich textual description and additional tags information, *e.g.*, genre, instrument, mood/theme tags, *etc*. The proposed method is evaluated using the MusicCaps (Agostinelli et al., 2023) benchmark, which consists of 5.5K expert-prepared music samples, each lasting ten seconds, and a genre-balanced subset containing 1K samples. To maintain fair comparison, objective metrics are reported on the unbalanced set, while qualitative evaluations and ablation studies are conducted on examples randomly sampled from the genre-balanced set.

**Evaluation Metrics.** In our quantitative evaluation, we employ both objective and subjective metrics. Objective evaluation encompasses three metrics: Fréchet Audio Distance (FAD)(Kilgour et al., 2019), Kullback-Leibler Divergence (KL)(Van Erven & Harremos, 2014), and CLAP score (CLAP)(Elizalde et al., 2023). FAD gauges audio plausibility, with a lower score indicating higher plausibility. KL-divergence measures the similarity between generated and reference music in terms of label probabilities, utilizing a state-of-the-art audio classifier trained on AudioSet(Gemmeke et al., 2017). A lower KL score suggests greater conceptual similarity. Additionally, the CLAP score quantifies audio-text alignment, using the official pre-trained CLAP model. For qualitative assessments, we follow the experimental design outlined in Copet et al. (Copet et al., 2023). Two aspects of the generated music are evaluated qualitatively: text-to-music quality (T2M-QLT) and alignment to the text input (T2M-ALI). Human raters assigned perceptual quality ratings on a scale of 1 to 100 in the text-to-music quality test. In the text-to-music alignment test, raters assessed the alignment between audio and text using the same rating scale.

### 5.1 COMPARISON WITH STATE-OF-THE-ARTS

As shown in Table 2, we compare the performance of JEN-1 with other state-of-the-art methods, including Riffusion (Forsgren & Martiros, 2022), and Mousai (Schneider et al., 2023), MusicLM (Agostinelli et al., 2023), MusicGen (Copet et al., 2023), Noise2Music (Huang et al., 2023a). These competing approaches were all trained on large-scale music datasets and demonstrated state-of-the-art music synthesis ability given diverse text prompts. To ensure a fair comparison, we evaluate the performance on the MusicCaps test set from both quantitative and qualitative aspects. Since the implementation is not publicly available, we utilize the MusicLM public API for our tests. And for Noise2Music, we only report the FAD score as mentioned in their original paper. Experimen-

Table 3: Ablation studies. From the baseline configuration, we incrementally modify the JEN-1 configuration to investigate the effect of each component.

| | QUANTITATIVE | | | QUALITATIVE | |
|---|---|---|---|---|---|
| CONFIGURATION | FAD↓ | KL↓ | CLAP↑ | T2M-QLT↑ | T2M-ALI↑ |
| baseline | 3.1 | 1.35 | 0.31 | 80.1 | 78.3 |
| + auto-regressive mode | 2.5 | 1.33 | 0.33 | 82.9 | 79.5 |
| + music in-painting task | 2.2 | **1.28** | 0.32 | 83.8 | 80.1 |
| + music continuation task | **2.0** | 1.29 | **0.33** | **85.7** | **82.8** |

tal results demonstrate that JEN-1 outperforms other competing baselines concerning both text-to-music quality and text-to-music alignment. Specifically, JEN-1 exhibits superior performance in terms of FAD and CLAP scores, outperforming the second-highest method Noise2Music and Music-Gen by a large margin. Regarding the human qualitative assessments, JEN-1 consistently achieves the best T2M-QLT and T2M-ALI scores. It is noteworthy that our JEN-1 is more computationally efficient with only 22.6% of MusicGEN (746M *vs.* 3.3B parameters) and 57.7% of Noise2Music (746M *vs.* 1.3B parameters).

## 5.2 PERFORMANCE ANALYSIS

**Ablation Studies.** To assess the effects of the omnidirectional diffusion model, we compare the different configurations, including the effect of model configuration and the effect of different multi-task objectives. All ablations are conducted on 1K genre-balanced samples, randomly selected from the held-out evaluation set. As illustrated in Table 3, the results demonstrate that i) JEN-1 incorporates the auto-regressive mode greatly benefiting the temporal consistency of generated music, leading to better music quality; ii) our proposed multi-task learning objectives, *i.e.*, text-guided music generation, music inpainting, and music-continuation, improve task generalization and consistently achieve better performance; iii) all these dedicated designs together lead to high-fidelity music generation without introducing much extra training cost.

**Generation Diversity.** Compared to transformer-based generation methods, diffusion models are notable for their generation diversity. To further investigate JEN-1's generation diversity and credibility, we provide identical textual prompts, such as descriptions involving general genres or instruments, to generate multiple different samples. As demonstrated on our demo page, JEN-1 showcases impressive diversity in its generation outputs while maintaining a consistently high level of quality.

**Generalization and Controllability.** Despite being trained with paired texts and music samples in a supervised learning manner, our method, JEN-1, demonstrates noteworthy zero-shot generation capability and effective controllability. Notwithstanding the challenges associated with generating high-quality audio from out-of-distribution prompts, JEN-1 still demonstrates its proficiency in producing compelling music samples. On our demo page, we present examples of creative zero-shot prompts, showcasing the model's successful generation of satisfactory quality music. Furthermore, we present generation examples as evidence of JEN-1's proficiency in capturing music-related semantics. Notably, our demo indicates that the generated music adequately reflects music concepts such as the genre, instrument, mood, speed, *etc.*.

## 6 CONCLUSION

In this work, we propose JEN-1, a text-to-music framework that directly models waveforms and integrates autoregressive and non-autoregressive training. Through multi-task objectives, JEN-1 achieves high-quality music generation from text descriptions. Extensive evaluations show JEN-1's superiority in quality, diversity, and controllability over strong baselines. This research advances the state-of-the-art in controllable text-to-music generation. Future directions include exploring hierarchical multi-stem music generation and external knowledge incorporation. We believe high-quality text-to-music generation will empower new creative workflows and reshape music composition and appreciation. As the field matures from research to practical applications, it bears great potential to augment human creativity.

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
