# OpenReview forum: "JEN-1: Text-Guided Universal Music Generation with Omnidirectional Diffusion Models"
_ICLR.cc/2024/Conference — ICLR 2024 Conference Withdrawn Submission_

### Official Review · Reviewer_Yv1M · 2023-10-17

**Soundness:** 2 fair
**Presentation:** 2 fair
**Contribution:** 2 fair
**Rating:** 3
**Confidence:** 4

**Summary:**

The paper proposes JEN-1, a novel text-to-music generation framework combining a masked autoencoder and an omnidirectional unet.  The main contributions are: 1. A masked autoencoder is proposed for robust audio reconstruction with a normalised latent space. 2. A modified unet is used for various tasks and performance boost. 3. A training strategy with different subtasks. The author also claim the proposed method achieve the state of the art performance.

**Strengths:**

1. This paper proposed the model achieved state-of-the-art results on music generation tasks.
2. The author's description of methods and experiments is mostly clear and accurate.
3. This paper replaces several components on the standard LDM structure, although some components used by the authors are not novel (e.g. masked autoencoder), it still has innovative significance in the field of this task (e.g. music generation).

**Weaknesses:**

1.The ablation studies in this paper are insufficient.
(a) As a key part of the paper, there is no experimental result validating the mask autoencoder in the experiments section. This makes the claims in Section 4.1 about its effects less credible.
(b) The meaning of each row in Table 3 is unclear. As state in 4.2, the AR mode should be proposed for the unified training tasks. And here AR mode appears with different training tasks.

2. I can only understand the author's use of the term unidirectional, but the AR nature of the omnidirectional structure is not clearly stated. The author should first clearly define the meaning of autoregressive (AR) before proposing the so-called omnidirectional structure with AR mode. It is not clearly explained how the model predicts $\epsilon_t$ when switching to AR mode. My understanding is UNET introduces causality through different padding and attention masks. However, this does not seem to be AR in a strict sense.

3. Recent research has shown that the performance of generative models heavily relies on the dataset used. The authors used a completely private dataset, which reduces the contribution of the model itself to the state-of-the-art performance.

4. Others: the authors did not specify the backbone network used to compute the FAD metric. Additionally, the authors did not write out the details of the diffusion model samplers used, such as the number of sampling steps, solver schemes, etc.

In summary, this paper is a bit over-claimed at present and lacks concrete experimental evidence.

**Questions:**

1. Enrich the ablation experiments, including the 1st-stage performance boosting for both reconstruction and generation.
2. Make the description of AR mode more clear.
3. Considering the importance of the dataset to generative tasks, more detailed information should be provided, such as scale, language, content overview, etc.
4. When using third-party evaluation metrics, specific implementation details should also be clearly annotated to make the results more reproducible.

**Details Of Ethics Concerns:**

The paper describes the training data in an overly vague manner without providing specific details of the dataset. This leads to doubts over whether there are copyright or licensing issues with the dataset.

---

### Official Review · Reviewer_TQzw · 2023-10-27

**Soundness:** 2 fair
**Presentation:** 1 poor
**Contribution:** 1 poor
**Rating:** 3
**Confidence:** 5

**Summary:**

The authors propose a latent diffusion diffusion model for music generation.
It consists in a waveform autoencoder and a diffusion model on latents.

They introduce a multi-objective loss and use for the diffusion Unet a causal or non-causal masking depending on the objective.

**Strengths:**

The introduction of noise in the latents during the autoencoder training phase seems novel and relevant.
The webpage features audio examples of high quality, which are on par with other similar methods.

The approach, except the multi-task objective and the two different modes (causal & non causal) of the models, is otherwise quite standard.

**Weaknesses:**

Overall, there is a certain lack of precision and details. There is no paragraph on:
- sampling method
- number of iteration
- generation time
- architectural details.
Code is likely to be omitted, the dataset is unknown.
As such it seems impossible, despite the nice-sounding results, to get reusable insights from this paper.


It is hard to understand the usefulness of the unidirectional diffusion model and how this affects sampling time. From Table 3, it seems that adding new tasks seems to improve

Lack of citations. Using an autoencoder with "non-discrete" latent variables on audio was made in 2021 in
RAVE: A variational autoencoder for fast and high-quality neural audio synthesis
https://arxiv.org/abs/2111.05011
This paper also features the dimensionality reduction + normalization of the latent space.

The use of an unknown "private" and undetailed dataset with "rich textual description" makes the comparison and evaluation of the contribution impossible.

**Questions:**

Since JEN-1 can "switch [from bidirectional mode] into unidirectional (autoregressive) mode", it means that the model is trained with both modes at the same time, so that the self-attention mask and padding change according to the current mode?

---

### Official Review · Reviewer_i8hT · 2023-10-30

**Soundness:** 2 fair
**Presentation:** 2 fair
**Contribution:** 2 fair
**Rating:** 3
**Confidence:** 4

**Summary:**

The authors propose JEN-1, a text-to-music diffusion generation model. The model uses a denoising diffusion approach on latent continuous embeddings generated from a neural audio codec, to synthesize 10-second music.

The neural audio codec was trained by the authors to encode and decode stereo signals at 48kHz. To train the audio codec autoencoder the authors use various tricks, such as masking parts of the embedding before decoding, normalizing the latent embedding, using a combination of reconstruction losses over the time and frequency domains, as well as a patch-based adversarial objective.

The diffusion model is based on a 1D Unet, using unidirectional and bidirectional convolutional and transformer layers. The diffusion model is trained on three tasks simultaneously: text-guided music generation, music inpainting, and music continuation. The authors use the Unet with unidirectional layers to train the music continuation, the bidirectional layers to train the music inpainting and both layer types to train text-guided music generation.

The authors compare the generated music from JEN-1 quantitatively and qualitatively to other models, and demonstrate superior performance. Furthermore, the authors demonstrate the effect of their design choices in an ablation study.

**Strengths:**

The paper is well presented and easy to read. The demo page and the provided samples are excellent. The ideas used to make the autoencoder audio codec work well for 48kHz stereo music are great. The paper compares the generated music to related work using relevant metrics. Furthermore, the combined training of different tasks makes for a compelling argument, demonstrated further by an ablation study that shows improvements across the board.

**Weaknesses:**

Unfortunately, the paper struggles from a lot of omitted details and statements that do not hold under scientific scrutiny. While the results sound very promising, the paper lacks evidence to support its claimed contributions. I don't think we can accept a paper that almost feels like the interesting details have been removed in order to keep fellow researchers from understanding and repeating the results.

In the following we list some of the issues.

In the first contribution the authors mention “JEN-1 utilizes an extremely efficient approach by directly modeling waveforms and avoids the conversion loss associated with spectrograms”, while I understand the point the authors are trying to make, the current statement is not true. JEN-1 requires the use of an audio codec to offload the encoding/decoding of music to latent embeddings, which is very similar to what related work has been doing (MusicLM with SoundStream, MusicGen with EnCodec, Stable Audio with VAE, etc.). Similarly to Stable Audio, JEN-1 also uses continuous latent representations. Furthermore, “extremely efficient” is a strong statement, the paper has no comparisons or ablations to substantiate such claims.

The authors argue that they use the continuous embeddings from their autoencoder since it gives better results than the quantized approach used by previous work (musicLM, musicgen). While this intuition might be true, there is no evidence provided to support this (neither through experimentation, nor through references). Furthermore, the authors state “This utilization of powerful autoencoder representations enables us to achieve a nearly optimal balance between complexity reduction and high-frequency detail preservation, leading to a significant improvement in music fidelity.” - This is a bold claim, nearly optimal balance - how do the authors know this balance is almost optimal? And how can they state a significant improvement in music fidelity?

In the second contribution the authors mention “JEN-1 integrates both autoregressive and non-autoregressive diffusion modes in an end-to-end manner [...]” A diffusion model is non-autoregressive in time since we are generating the entire sequence after N denoising steps, irrespective of the type of layers used (uni/bidirectional). If there has been related work on this, please cite it. As far as I understood, the authors claim to have an autoregressive diffusion model because of the unidirectional convolutional and transformer layers. If this is the case, I would disagree with this statement.

The paper would benefit from an extensive ablation study separating the audio codec from the diffusion model, at this time it is unclear which component contributes how much to the final performance. For example, using EnCodec and Descript Audio Codec as audio encoders/decoders and training the authors’ proposed diffusion model on their latent music representations would demonstrate the effect of the proposed audio codec.

The paper could greatly benefit from an architecture diagram visualizing how all components connect to each other. Many of the questions asked below would be answered with such a figure.

There is very little information on the setting and methodology used to collect the qualitative results with human subjects. (Though I agree that the results sound good, so I don't question the results per se.)

Apart from the information given in the paper, which does not suffice to reconstruct the provided results, there seems to be little effort made regarding reproducibility. It would be highly beneficial for the research community and the reproducibility of this work if the corresponding code (and music copyrights permitting, model weights) were open-sourced.

Finally, even though the results sound very good, it is unclear whether they were selected randomly or cherry-picked. Furthermore, the authors mention JEN-1 has autoregressive capabilities, but do not demonstrate this on a prolonged generation (compare with generating for 30+ seconds as demonstrated by recent autoregressive models such as musicLM or musicgen).

Generally, the paper needs more detail. If the authors can explain these weaknesses, I will increase my score.

**Questions:**

Is the audio codec trained separately and then frozen while the diffusion model is trained? This is what I expect. In case the authors manage to train the audio codec and diffusion model end-to-end simultaneously, I will revise my review.

Can the authors give more information on the losses used to train the autoencoder audio codec? There are many important details omitted from “which consists of an autoencoder trained by a combination of a reconstruction loss over both time and frequency domains and a patch-based adversarial objective operating at different resolutions.” - if the authors are using the same setup as EnCodec, SoundStream, or Descript Audio Codec then please clarify this, otherwise more information on the setup is needed.

Is the conditioning information passed via channel or via cross-attention? I assume only the mask is passed via channel and the Flan-T5 text embedding is passed via cross-attention? Section 4.2 states “[...] the input consists of the noisy sample denoted as $x_t$, which is stacked with additional conditional information” - what is this additional conditional information? The masks for inpainting and continuation tasks?

In Section 4.2 Task Generalization via In-context Learning, the authors mention “apart from the original latent channels, the U-Net block has 129 additional input channels (128 for the encoded masked audio and 1 for the mask itself).” - how many channels does the Unet have in total? Does the Unet have 129 channels? It is never specified what the exact dimensions of the latent representation are (variable c in Section 4.1), so it is unclear what size of tensor is being diffused by the model.

In Section 4.1 Normalizing Latent Embedding Space, the authors mention “Additionally, we incorporate a dimension reduction strategy with 5% least important channels to enhance the whitening process further and improve the overall effectiveness of our approach.“ - What is meant by dimension reduction with 5% least important channels? How is this done?

Can the authors further explain how their unidirectional layers represent an autoregressive diffusion model? Since the authors do not cite any related work on this, I assume that this is an original contribution? Nevertheless, please provide experiments or theoretical formulations demonstrating how using unidirectional layers in a diffusion model allows it to become an auto-regressive model.

Can the authors give more technical details on how the ablation study was conducted? Is the model trained from scratch for each entry in the table? What about the batch splitting described in Implementation Details?

Regarding qualitative evaluations, could the authors give more information? Did human raters follow a MUSHRA or MOS test, or something else? How many raters were involved, how many samples did each human rate, etc.?

Finally, can the authors give more details on the data used to train the model? Where does it come from? How were the captions sourced/created? What exact information is contained in the captions?

**Details Of Ethics Concerns:**

The authors use 5’000 hours of music with corresponding textual descriptions to train JEN-1. No additional information on this data is given and whether the authors have the rights to use this data.

In Section 4.3 Music Inpainting Task, the authors mention “[...] eliminate undesired elements like noise and watermarks from musical compositions”. The reason for watermarking is to discourage users from misusing copyrighted material; so the authors are suggesting to remove watermarks without the right holder’s consent?

---

### Official Review · Reviewer_seXG · 2023-11-01

**Soundness:** 2 fair
**Presentation:** 3 good
**Contribution:** 3 good
**Rating:** 6
**Confidence:** 3

**Summary:**

The paper presents a comprehensive model capable of text-to-music generation, inpainting, and continuation tasks. Central to this endeavor, the authors have introduced a distinctively masked autoencoder incorporating SVD-based normalization tailored for converting stereo 48kHz audio to a latent space. The diffusion backbone of this model is supported by a novel 1D Conv-based Unet, which is trained for both causal and non-causal generation. A mixed training scheme is used, which is mixed with unmasked generation, inpainting, and continuation. The results show that the proposed model generates music with quality in both quantitative and qualitative metrics.

**Strengths:**

1. A novel masked autoencoder with SVD-based normalization that supports generating 48kHz stereo audio.
2. A novel 1D Unet that supports both causal and non-causal generation
3. The paper is well-written. The generated audio quality is good.

**Weaknesses:**

Although I think the paper is well-organized and there are sufficient contributions, I would raise a concern about the overclaim of "mixed task". The JEN-1 is trained with explicit in-painting and continuation tasks, which is valid to justify this model being "multi-task." However, diffusion-based model such as AudioLDM and Noise2Music also supports inpainting and continuation (as a special case of continuation) due to the nature of diffusion. I think clarifying that would be more rigorous. I would be happy to raise my score if the author would clarify more on this issue.

**Questions:**

Would this model support online or streaming generation as it supports causal generation?